# Software Assisted Multi-Tiered Mass Spectrometry Identification of Compounds in Traditional Chinese Medicine: *Dalbergia odorifera* as an Example

**DOI:** 10.3390/molecules27072333

**Published:** 2022-04-04

**Authors:** Mengyuan Wang, Changliang Yao, Jiayuan Li, Xuemei Wei, Meng Xu, Yong Huang, Quanxi Mei, De-an Guo

**Affiliations:** 1National Engineering Research Center of TCM Standardization Technology, Shanghai Institute of Materia Medica, Chinese Academy of Sciences, 501 Haike Road, Shanghai 201203, China; s19-wangmengyuan@simm.ac.cn (M.W.); clyao@simm.ac.cn (C.Y.); lijiayuan@simm.ac.cn (J.L.); s19-weixuemei@simm.ac.cn (X.W.); xumeng2@simm.ac.cn (M.X.); hydess@simm.ac.cn (Y.H.); 2School of Pharmaceutical Sciences, University of Chinese Academy of Sciences, No. 19A Yuquan Road, Beijing 100049, China; 3Shenzhen Baoan Authentic TCM Therapy Hospital, Shenzhen 518101, China; quanxi-mei@126.com

**Keywords:** software, mass spectrometry, flavonoid, *Dalbergia odorifera*

## Abstract

The complexity of metabolites in traditional Chinese medicine (TCM) hinders the comprehensive profiling and accurate identification of metabolites. In this study, an approach that integrates enhanced column separation, mass spectrometry post-processing and result verification was proposed and applied in the identification of flavonoids in *Dalbergia odorifera*. Firstly, column chromatography fractionation, followed by liquid chromatography–tandem mass spectrometry was used for systematic separation and detection. Secondly, a three-level data post-processing method was applied to the identification of flavonoids. Finally, fragmentation rules were used to verify the flavonoid compounds. As a result, a total of 197 flavonoids were characterized in *D. odorifera*, among which seven compounds were unambiguously identified in level 1, 80 compounds were tentatively identified by MS-DIAL and Compound Discoverer in level 2a, 95 compounds were annotated by Compound discoverer and Peogenesis QI in level 2b, and 15 compounds were exclusively annotated by using SIRIUS software in level 3. This study provides an approach for the rapid and efficient identification of the majority of components in herbal medicines.

## 1. Introduction

Traditional Chinese medicine (TCM) plays an important role in the treatment and prevention of diseases within China and the world at large. To remove the mask and achieve the scientific utilization of TCM, ongoing devotions have been made to probe the chemical basis, metabolism and mechanism of action, among which the chemical basis is a cornerstone [1,2,3]. However, due to the complexity of intrinsic metabolites, the in-depth identification of TCM components still remains to be a bottleneck. Moreover, unknowns in complex matrices obtained by phytochemical isolation and purification can be unambiguously identified by nuclear magnetic resonance [4]. The low-throughput plus tedious isolation process of this method confines its application for a holistic and fast analysis of TCM components. In the past decades, due to its unprecedented selectivity and sensitivity, liquid chromatography–tandem mass spectrometry (LC–MS), especially multistage high-resolution MS (HRMS), has become increasingly popular and a most widely used technique for the separation and identification of TCM components [5,6,7]. In addition to the masses of the targets, HRMS/MS can convey certain structural information through abundant spectral fragments, which are critical for metabolites identification. However, manual interpretations and annotations of the MS fragments in an experience-dependent manner are far behind analytical requirements and other cutting-edge techniques, such as proteomics [8] and genomics [9], and often lead to inconsistent results from different laboratories. These obstacles hinder MS from becoming a gold standard for TCM analysis and other multiple applications.

In recent years, multiple kinds of MS software have been developed to standardize and accelerate the identification procedures, especially in the field of metabolomics [10,11,12]. Moreover, most of the TCM analysis studies can be categorized into plant metabolomics. According to Metabolomics Standards Initiative (MSI) [13], metabolite annotation can be generally categorized into four different levels. An upgraded criteria proposed by Schymanski [14] provides cleaner and more practical classification. Five levels are included, and level 1 is relative to confirmed structure. Level 2 refers to probable structures identified by library spectrum match (2a) and diagnostic evidence (2b). Level 3 involves multiple tentative candidates of isomers, but not one exact structure (such as positional isomers). Levels 4 and 5 provide only mass formula or exact mass. Since level 1 identifications are dependent on reference standards, many software tools have been developed to promote compounds in subsequent levels to level 2, or another higher level. The related freely available or commercial software has been adopted in a few researches of herbal analysis [15,16,17]. A self-built database with 642 potential compounds was imported into UNIFI to assist the multicomponent characterization of Shuanghe decoction [18]. A specific tool was developed with Compound Discoverer for the semi-automatic identification of fiavonoids, anthocyanins, ellagitanntns, proanthocyanidins and phenolic acids, leading to 147 identified compounds above level 3. Isotopic pattern filter in Compound Discoverer was applied for screening 12 *S*-containing compounds in medicinal used *Pueraria* species [19]. By extracting the Mass2Motifs (co-occurring fragments and neutral losses) from the spectral data, analyzing by MS2LDA workflow in combination with mass spectral molecular networking and in silico fragmentation prediction, 124 compounds from *Pheretima aspergillum* (Di-long) were effectively characterized in Global Natural Products Social Molecular Networking (GNPS) and UNIFI platforms [20]. In spite of some successful application cases, it is still obscure how to maximally increase identification confidence with existing software tools.

*Dalbergia odorifera* T. Chen is a famous Chinese herbal medicine native to Hainan Province in Southern China. Phytochemical studies have shown that the main active components of *D. odorifera* are volatile oils and flavonoids [21]. Reports showed that flavonoids from *D. odorifera* have good therapeutic effects on cardiovascular diseases, blood diseases and other inflammation-related diseases [22]. Therefore, flavonoids are expected to be the quality index to evaluate *D. odorifera*. However, the current quality control of *D. odorifera* only focuses on the total content of volatile oils [23], which cannot fully reflect the component-efficacy relationship, and seriously limits the in-depth research and clinical application. Therefore, it is in urgent need to systematically characterize the flavonoids in *D.*
*odorifera*.

In this study, multiple intelligent data post-processing tools were adopted for accurate and efficient identification of flavonoids in *D. odorifera*. According to Schymanski’s criteria, level 1 identification was acquired by comparing with reference standards, level 2a identification was achieved by searching spectral databases, level 2b identification was realized by matching the structural databases and in silico fragments and level 3 identification was carried out by de novo structural elucidation (Figure 1). Different tools were compared for their identification performances. In addition, the compounds displaying a high similarity score in level 2a and other flavonoids in the mzCloud database were defined as quasi reference standards. Fragmentation rules were elucidated from reference standards, along with quasi reference standards, to validate the annotation accuracy in levels 2a–3. As a result, 197 compounds in *D. odorifera* were annotated in level 3 or higher levels. It is anticipated that this study could provide a valuable reference for the rapid and efficient identification of the overall components in herbal medicines with bioinformatics tools.

## 2. Results and Discussion

At the first step, the extract was fractionated by column chromatography before LC–MS detection. This step makes it possible to detect more components in *D. odorifera* by increasing the peak capacity, reducing the interference of co-eluting components and enriching the minor components [24]. Subsequently, reference substances and three in silico data-processing solutions were used to improve the reliability and efficiency of the identification [25]. Following Schymanski rules, the identification results were divided into three levels (1, 2a, 2b and 3), according to the confidence and the analysis mechanism. Finally, the identification results were confirmed by manual interpretation of the fragments in levels 2a–3.

### 2.1. Column Chromatography Fractionation

In this section, the extract was segmented into eight fractions (Fr1~Fr8) by silica gel, and a total of 3456 mass features were detected, with 666, 719, 754, 647, 591, 534, 293 and 281 mass features in Fr1~Fr8, respectively. Compared with 851 mass features detected in the total extract, silica gel column fraction enhanced chromatographic separation and improved sensitivity. In this experiment, we mainly characterized small molecule flavonoids, and a total of 197 flavonoids were annotated (Appendix A). In addition to monoflavones, a large number of compounds with molecular weights within 500–800 Da in Fr1~Fr2 were also detected. These compounds were deduced to be biflavones and flavonoids with fatty acid side chains, which were not discussed in detail.

### 2.2. Level 1: Identified by Comparing with Reference Standards

To identify known compounds, all the available reference standards of flavonoids which potentially exist in *D. odorifera* were collected in the author’s laboratory. Independent and orthogonal data, including retention time, accurate mass and MS/MS, were acquired for these compounds under identical experimental conditions [26]. The candidates were filtered according to the following rules: (1) a retention time error not exceeding 0.05 min, (2) an exact mass error not exceeding 5 mDa and (3) matching fragment ions. Seven compounds (F_12_, F_13_, F_14_, F_15,_ F_20,_ F_22_ and F_24_) were identified by using this method, and their structures are shown in Figure 2. Albeit unambiguous identification of the targets, it is difficult to obtain enough reference materials for extensive compound identification, due to high price of reference substances, difficulty of collection and lack of reference substances for some compounds. Therefore, the remaining 190 compounds were performed with the following level 2a identification.

### 2.3. Level 2a: Searching in Spectral Databases

Given the targets without chemical reference standards, searching the public or commercially available spectral databases acts as a supplementary approach. At present, the common databases, containing fragmentation spectrum information of small molecules from different sources, mainly include METLIN (https://metlin.scripps.edu/, accessed on 5 February 2022), MassBank (http://www.massbank.jp, accessed on 5 February 2022), mzCloud (https://www.mzcloud.org/, accessed on 5 February 2022) and so on. There are also a small number of plant-specific databases, such as ReSpect [27] and PlaSMA [28]. A total of 2957 flavonoids were included in ReSpect, which is a part of the MS-DIAL database. METLIN reportedly has more than 500,000 molecular standard data with fragmentary spectra. Massbank records 89,826 unique spectra, 15,059 unique compounds and 16,840 unique isomers. Since mzCloud contains the database of high-resolution and low-resolution MS^n^ spectrum obtained under many experimental conditions, it solves the problem of spectral reproducibility to a certain extent.

To enable batch processing and the visualization of results, database searching can be implemented by using software, such as MS DIAL, Compound Discoverer (CD), RAM, etc. Database searching is usually accomplished by calculating the total score integrating MS1 similarity and MS/MS similarity, and RT similarity in some cases. For the MS/MS spectral similarity, the “dot product” scoring method is often adopted to perform spectral matching [29]. A high score of spectral database matching often indicates high confidence in structural annotations. Recently, a method called entropy similarity that outperformed dot product similarity was proposed [30].

Although a spectral database search is a simple and efficient approach for metabolite identification, insufficient MS/MS spectra from authentic compounds in databases and differences in mass spectrometry instruments are the main limiting factors for this approach. Moreover, components from herbal medicines contributed to only a small proportion of these databases. In addition, positional isomers are widespread in herbal medicines, and they often display closely similar mass spectra. It is difficult to distinguish isomers with similar structures solely by this method. To overcome this problem, complementing other orthogonal information, such as retention time, collision cross-section and biosynthesis pathways, will lead to a higher identification confidence. Here, based on software of MS DIAL 4.70 and CD 3.3, we identified 80 flavonoids in *D. odorifera* by searching spectral databases.

#### 2.3.1. MS DIAL

MS DIAL is an untargeted metabolomics data processing pipeline that is suitable for both data-dependent and data-independent data [31]. According to the latest statistics, the MS-DIAL database, integrating publicly available databases (such as MassBank and GNPS), contains 13,303 unique compounds in positive ion mode and 12,879 unique compounds in negative ion mode. Here, we identified the flavonoids in *D. odorifera* by searching the positive spectral database. Moreover, the identification score cutoff was set at 80%, a high confidence in structure annotation. Finally, a total of 50 flavonoids from *D. odorifera* were effectively characterized.

Compound **104** (Rt = 24.13 min) displayed a precursor ion [M + H]^+^ at *m*/*z* 269.0794, matching the molecular formula [C_16_H_13_O_4_]^+^. By comparing the MS/MS spectrum with positive spectral database, it was identified as 7-hydroxy-3-(4-methoxyphenyl)-4*H*-chromen-4-one (Biochanin B) with a total score at 96.6%. As shown in Figure 3a, the fragmentation spectrum of compound **104** highly matched with the reference spectrum in fragmentation and peak intensity. Furthermore, the instrument of the reference spectrum was the same as our instrument, thus further illustrating the reliability of the result. Correspondingly, compound **68** was also assigned to Biochanin B, with a lower similarity of 86.40%. However, the fragmentation spectrum of compound **68** was visually different from the reference spectrum. As shown in Figure 3b, the product ions at *m*/*z* 226.062 and 197.060 were well matched in the query and reference spectrum, while some product ions (*m*/*z* 253.049, 237.054, 213.091, 181.065, 170.072 and 156.057) show different relative peak intensities, and some product ions (*m*/*z* 137.032, 133.056, 118.041 and 108.02) are in very low abundance. The product ions at 137.032, 133.056 and 118.041 in compound **104** match the molecular formula [C_7_H_5_O_3_]^+^, [C_9_H_9_O]^+^ and [C_8_H_6_O]^+^, corresponding to ^1,3^A, ^1,3^B and [^1,3^B-CH_3_]^+^ ions, indicating that compound **68** has no methoxy group in the B ring (Figure 3c). In summary, it was speculated that compound **68** was also an isoflavone, a positional isomerism of **104**.

Compound **89** (Rt = 19.30 min), exhibiting the molecular ion [M + H]^+^ at *m*/*z* 301.0698, was automatically labeled as (2*Z*)-4,6-dihydroxy-2-[(3-hydroxy-4-methoxyphenyl)methylidene] -1-benzofuran-3-one. Appendix A shows that the fragments of compound **89** are highly matched with the reference spectrum, but the relative intensities of corresponding ions are quite different, with low a similarity score of 80.6%. It may be due to the use of different instruments or the great difference in fragmentation energy. Such structure annotation results may have certain deviations, because the correct structure may be isomers that are very relevant to the annotated structure. By taking this into account, we compared the fragment spectra of compound **89** collected under different instruments and different energies (Appendix A). The results showed that the base peak ions were always at *m*/*z* 229.0499, although the relative intensity of some ions varied greatly. The discrepancy indicates the possible wrong automatic identification of compound **89**. Therefore, careful manual inspections are usually necessary in similarity, matching of the spectra. In addition, mass spectrometry fragments of samples with different concentrations were compared, and no significant difference was observed with consistent identification results.

#### 2.3.2. Compound Discoverer (CD) Based on mzCloud Database

CD software can make full use of the rich high-resolution accurate-mass (HRAM) data produced by an Orbitrap mass spectrometer to annotate the structures of small molecule metabolites. In addition to spectral database exact searching, similarity searching of mzCloud and automated annotation of spectra with predicted fragments enable CD software to identify more metabolites [32]. Here, 63 components were successfully identified.

By comparing the MS/MS spectrum with mzCloud database, compound **147** was annotated as Biochanin A, of which the mzCloud Best Match is 98.2%. The mirror plot of the mzCloud search result of compound **147** indicates confident identification. As shown in Appendix A, some matched fragments were automatically annotated with accurate masses, structural fragments and charge states, and this is an improvement compared with MS-DIAL. For compound **147**, the product ions 269.0455, 241.0507, 213.0555 and 153.0189 are annotated as [C_15_H_9_O_5_]^+^, [C_14_H_9_O_4_]^+^, [C_13_H_9_O_3_]^+^ and [C_7_H_5_O_4_]^+^, respectively, which helps to infer the structure of questions. In this way, even if some product ions do not match well with the reference spectrum, the structures could be elucidated from the annotated product ions.

### 2.4. Level 2b: Searching in Molecular-Structure Databases

Compared with the spectral database containing only a few compounds, the molecular-structure databases are more comprehensive. Molecular-structure databases commonly used in the identification of small molecule metabolites include ChemSpider (http://www.chemspider.com/, accessed on 7 February 2022), PubChem (https://pubchem.ncbi.nlm.nih.gov/, accessed on 7 February 2022), LipidMAPS (http://www.lipidmaps.org/, accessed on 7 February 2022), DrugBank (https://www.drugbank.ca/, accessed on 7 February 2022), FooDB (http://foodb.ca/, accessed on 7 February 2022), etc. Different databases have their own characteristics, which may be different in the types of compounds contained, in the records of compound information, etc.

Theoretical fragmentation can be obtained in software, such as Progenesis QI (MetFrag algorithm), to rank the potential targets. MetFrag [33] takes a target compound structure and iteratively breaks bonds to produce a series of possible fragments, which are then compared to each ion in acquired MS/MS spectra. However, because the molecular structure databases contain a huge number of closely similar structures, and their theoretical fragments are independent of experimental data, this often leads to ambiguous identifications. As for a certain group of compounds, fragmentation rules can be elucidated from experimental data; therefore, in silico MS/MS spectra can be achieved from chemical structures. Databases, such as LipidBlast [34] and Acyl-CoA libraries, have been successfully constructed and applied for the identification of some sequential constituents. These libraries greatly improved the identification confidence and provide quasi-level 2 results.

In this work, two kinds of software with different approaches of searching in molecular structure databases were selected for the structural annotation of flavonoids in *D. odorifera*. As a result, 95 compounds were identified.

#### 2.4.1. Progenesis QI

Progenesis QI is a small molecule discovery analysis vehicle for processing metabolic profiling data. It is worth noting that, on top of the commercial databases, Progenesis QI software also supports users when they define their own structure databases for compound characterization [35]. Here, we identify the flavonoids in *D. odorifera* by searching four molecular-structure databases with Progenesis QI. Two online databases, HMDB and Natural Products (NPDB), and the other two self-built databases, namely a structure database of *D. odorifera* components obtained from Reaxys (DODB, 112 components) and a theoretical structure database of flavones (TSDB, 40,095 components), were consulted, respectively. As a result, 157 components were annotated in HMDB, 177 components in NPDB, 175 components in DODB and 178 components in the TSDB. Although the number of annotated components in different databases varied slightly, they worked out in a similar way. For instance, compound **58** (Rt = 15.05 min) exhibited a precursor ion at *m*/*z* 269.0806 and was characterized as formononetin in NPDB, DODB and TSDB. In addition, for those isomers with similar structures, they usually produce similar fragmentation spectra, causing equivocal annotations. As shown in Figure 4a, compounds **58**, **104** and **177** generated the same product ions at *m*/*z* 226.0635, 197.0607, 181.0658 and 169.0656, causing them all to be characterized as formononetin. However, as one can see from Figure 4a, the product ions of *m*/*z* 253.0608, 226.0635, 181.0657, 137.0239 and 118.0418 vary greatly in different compounds, indicating the differences between their structures. In summary, it can be concluded that searching in structural databases can only distinguish compounds with great structural differences.

#### 2.4.2. Compound Discoverer (CD) Based on ChemSpider Database

In addition to an online search of the mzCloud spectral database, CD can also annotate compound structures by searching Chemspider. The potential structure candidates generated by the ChemSpider search were ranked by comparing the experimental fragments to the extensive, fully curated MS^n^ fragment ions in the mzCloud mass spectral library [36]. In this work, 150 components were successfully annotated in *D. odorifera* by searching in ChemSpider.

Compounds **141** and **168** displaying a precursor ion [M + H]^+^ at *m*/*z* 285.1112 were annotated as flavanone and chalcone, respectively. As illustrated in Figure 4b, compound **141** generated main product ions at *m*/*z* 181.0657, 131.0497, 165.0706, 107.0494 and 210.0684. Compound **168** yielded abundant product ions at *m*/*z* 123.0445, 131.0497, 105.0701 and 147.0447. The significant differences in the fragmentation spectraof compounds **141** and **168** suggested that their structures were different. However, the relative intensity changes of the product ions in the spectra are often ignored when annotating compounds from structural databases, and this is usually the main reason for the wrong annotation of compounds. Figure 4c shows that compound **167** and compound **194** generate the same product ions at *m*/*z* 255.0663, 210.0684, 199.0763, 181.0656, 168.0578 and 157.0655, whereas there was a visibly striking difference in the peak relative response of these product ions. Hence, these two compounds were tentatively annotated as the same compound. From this result, we can clearly see the drawbacks of the structural database search. In addition, multiple compounds with similar structures may be obtained simultaneously when searching in structure databases, which is more conducive to the classification of compound structures. For instance, according to the results of the top one of the candidate compound ranking, compound **27** was identified as a flavonoid. As shown in Appendix A, the top five candidates are flavonols, and the only differences in their structures are the positions of hydroxyl group and methoxy group.

#### 2.4.3. Level 3: De Novo Molecular Structure Identification

Continuous efforts have been made to explore the chemical basis of herbal medicines, and there are still plenty of unreported compounds. These brand-new compounds can be annotated by de novo molecular structure elucidation. De novo molecular design is a process of automatically proposing new chemical structures to optimally meet the desired molecular profile [37]. In the process of structure annotation, the de novo analysis method does not rely on the spectral database, but reconstructs the structure of compounds according to the distribution information of fragmentation spectrum of unknown compounds. In recent years, de novo technology has been widely used in protein or peptide structure reconstruction, but the application in the identification of small molecular compounds is relatively scare. In this paper, we introduce a method developed by Sebastian Böcker’s team [38] combining structural fingerprint with fragmentation trees (implemented by SIRIUS 4.9.8 software) for the de novo identification of unknown compounds in *D. odorifera*.

Mass spectrometry fragments are produced by the fragmentation of precursor ions; therefore, all the molecular formulas of mass spectrometry fragments are theoretically the substructures of precursor ions. The fragment tree consists of nodes, corresponding to precursors and fragments, and the entire fragmentation tree displays the predicted multistage mass spectrogram. The structural fingerprint predicted from fragmentation tree and fragment spectrum contains the fragment features and intensity information of fragmentation spectrum [39]. For the unknowns, SIRIUS can perform de novo structure elucidation by predicting molecular fingerprints, or annotate them by searching theoretical databases. Here, we annotate a total of 197 components in *D. odorifera* through SIRIUS software, 15 of which were used for the results in level 3.

When SIRIUS is used to identify unknowns, firstly, the software will generate a fragment tree with the highest confidence, based on the fragmentation spectrum of queries. From the fragment tree, the relative intensity and neutral loss information of product ions can be easily obtained (Appendix A), and these greatly enrich our cognition of the mass spectrometry fragmentation behavior of unknowns and make product ion annotation more convenient. As shown in Appendix A, compound **138** exhibited the molecular ion [M + H]^+^ at *m*/*z* 257.0803 and generated product ions at *m*/*z* 153.0188, 103.0544 and 131.0496, corresponding to [M-C_8_H_8_]^+^, [M-C_4_H_4_O_2_-C_2_H_2_O-CO]^+^ and [M-C_4_H_4_O_2_-C_2_H_2_O]^+^, respectively. It can be easily inferred that product ions 153.0188 and 103.0544 correspond to fragments ^1,3^A (C_7_H_4_O_4_) and ^1,3^B (C_8_H_6_), respectively, and the compound has two hydroxyl substituents in the A ring and no substituent in the B ring. Furthermore, according to the fragment tree, the SIRIUS software predicts a corresponding molecular fingerprint with the CSI: finger ID. For compounds existing in the molecular structure database, this predicted molecular fingerprint can be used to search for compounds with similar structures in the database. For novel compounds that have not been reported, this predicted fingerprint supports de novo structure elucidation or searching for similar structures from theoretical databases [38].

### 2.5. Result Validation

In order to verify the identification results in levels 2a–3, the fragmentation spectra of flavonoids were selected for the study of cracking rules. Traditionally, only reference compounds were adopted for the elucidation of cracking rules. However, due to the limit of the reference standards (a), the rules are usually fragmentary and incomplete. In this study, compounds with a score higher than 90% in level 2 (b) and other rare flavonoids in mzCloud database (c) (fragment spectra were collected in HCD mode) were ticketed for the quasi reference compound. Therefore, cracking rules were elucidated from 39 (quasi) reference compounds 2.3 times more than the traditional method. The 39 compounds include 12 types of flavonoids, namely four flavones, six flavonols, nine isoflavones, one aurone, six flavanones, two chalcones, two flavanols, four neoflavones Ⅰ, one neoflavonoid Ⅱ, one neoflavonoid Ⅲ, two neoflavonoids Ⅳ and one neoflavonoid Ⅷ. The detailed structures are shown in Figure 2 and Appendix A.

#### 2.5.1. Flavonoids and Flavonols

By analyzing the MS spectra of F1–F4 (flavonoids), it can be demonstrated that, in the HCD fragmentation mode, flavonoids usually retain the parent ions ([M + H]^+^/[M − CH_3_ + H]^+^) with higher abundance. The common abundant fragments of flavonoids are ^1,3^A, ^1,4^B, ^0,4^B and ^0,2^B. Furthermore, characteristic [^1,3^A + H_2_O + H]^+^ is often observed. In addition, flavonoids can lose the neutral molecule CO (28 Da) at position 4 to generate strong [M-CO + H]^+^/[M-CH_3_-CO + H]^+^ fragments. Our analysis of the MS spectra of compounds F_5_-F_10_ showed that the flavonols are also prone to ^1,3^A and ^0,2^B in HCD mode. In addition, [^1,3^A + H]^+^ usually acts as the base peak ion, such as *m*/*z* 153.0186 of F_5_ and *m*/*z* 137.0233 of F_10_. In addition, flavonols can generate characteristic ^0,3^A fragments, as well as [M-H_2_O + H]^+^ and [M-H_2_O-nCO]^+^ fragments, due to the presence of the hydroxyl group at the 3-position (Appendix A). Accordingly, compounds **31**, **46** and **86** identified by SIRUS were substantiated.

#### 2.5.2. Isoflavones

According to compounds F_11_–F_19_, the common fragmentation mode of isoflavones is ^1,3^X fragmentation, and strong [^1,3^A + H]^+^ and [^1,3^B + H]^+^ can be observed in the secondary spectrum, such as *m*/*z* 137.02 and 119.05 of F_14_. In addition, a series of peak ions showing the loss of neutral molecule CO (28Da) are detected in the spectrum, such as *m*/*z* 243.06, 215.07 and 159.04 of F_12_, which are characteristic for isoflavones (Appendix A). Compounds **61**, **98** and **113** conformed to the above rules.

#### 2.5.3. Flavanones, Chalcones and Flavanols

The MS/MS of compounds F_20_–F_27_ indicated that both the flavanones and chalcones showed main fragments at ^1,3^X and ^1,4^X, and the loss of ring B ([M-B + H]^+^). The base peak ion is generally [^1,3^A + H]^+^, contributing more than 40% of the total intensity of all fragments, which is the most prominent feature of flavanones and chalcones. In addition, [^1,3^B + H]^+^ and [^1,4^B + H]^+^ are second only to [^1,3^A + H]^+^ in intensity (Appendix A). It is generally difficult to distinguish between flavanones and chalcones only by using an MS/MS spectrum. However, according to literature reports [40], due to the existence of the double bond at the C2–C3 position of chalcones, the conjugated structure displays a different UV spectrum from that of flavanones.

Our analysis of F_28_ and F_29_ showed that the flavanols mainly fragment around 3-OH. Moreover, the common fragments are ^1,3^A, ^0,2^A, ^0,3^A and ^2,4^X. Due to the presence of 3-OH, flavanols are prone to produce dehydrated fragments during fragmentation, such as [^0,2^A-H_2_O + H]^+^ and [^1,3^B-H_2_O + H]^+^ (Appendix A).

#### 2.5.4. Neoflavonoids Ⅰ

By analyzing F_30_–F_33_, it can be observed that, due to the existence of epoxy bonds, the fragmentation sites of the neoflavonoids Ⅰ are in the B ring and the C ring, and the main fragments are ^5,10a^D, ^5,11a^D, ^6,10a^A and ^6,11a^A (Figure 5a). Affected by the 5- and 11-position oxygen atoms, the neoflavonoids Ⅰ are prone to produce high-abundance [^6,11a^A + H]^+^ fragments, such as the *m*/*z* 137.0589 produced by F_30_, which is expected to be a key to characterize such compounds. Compounds **145** and **150** were identified as neoflavonoids Ⅰ, and the abundant ions 152.0472 and 123.0444 in the MS/MS spectra corresponded to [^6,11a^A + H]^+^ fragments, and the identification results were verified.

#### 2.5.5. Neoflavonoids Ⅱ

The fragmentation behavior of neoflavonoids Ⅱ was predicted from compound F_34_. As shown in Figure 5b, F_34_ is prone to neutral loss of CO and 2CO to obtain the abundant ion [M-CO + H]^+^ (*m*/*z* 241.0495) and [M-2CO + H]^+^ (*m*/*z* 213.0546). In addition, due to the existence of ester bonds, ring B is prone to ring opening in mass spectrometry. For example, the abundant ion at *m*/*z* 157.0648 is obtained after ring A and ring B fragmentation. In contrast, ring C is relatively stable and generally does not open.

#### 2.5.6. Aurones

As shown in Figure 6a, compound F_35_ produced high-abundance complementary ions at *m*/*z* 134.04 and 137.03 corresponding to ^1,3^B and ^1,3^A, respectively. The former is base peak ion and a radical ion, which may be its characteristic fragment. In addition, F_35_ generates the [M-CH_3_ + H]^+^ fragment by losing the neutral molecule CH_3_ (15 Da) and generates the [^1,4^A + H]^+^ fragment.

#### 2.5.7. Neoflavonoids Ⅲ

The fragmentation behavior of the neoflavonoids Ⅲ was predicted from compound F_36_. As shown in Figure 6b, the main MS/MS fragments of F_36_ were ^1,3^AB, ^1,4^ Band ^0,3^X, where [^1,3^A + H]^+^ (*m*/*z* 123.04) is the base peak ion. Since there is no C4 carbonyl group, the [^1,3^A + H]^+^ fragment is one degree larger in unsaturation than other flavonoids, and this may be its characteristic ion.

#### 2.5.8. Neoflavonoids Ⅳ

By analyzing the MS spectra of F_37_ and F_38_, it was found that, due to the C1 epoxy and the C3 hydroxyl group, the neoflavonoids Ⅳ are prone to the cleavage of the single bond between C1and C2. Therefore, the common fragments are ^1,2^ B, ^0,2^ B and B^+^. The base peak ion is generally [^1,2^B + H]^+^, and its abundance is much higher than other ions, which is the most characteristic fragmentation mode of neoflavonoids Ⅳ (Figure 6c). For example, compound **70** was attributed to neoflavonoid Ⅳ, and its base peak ion was *m*/*z* 138.0318, corresponding to [^1,2^B + H]^+^, demonstrating the identification reliability.

#### 2.5.9. Neoflavonoids Ⅷ

As shown in Figure 6d, F_39_ fragmentation mainly involves the opening of rings D and E. For example, rings D and E of compound F_39_ are cleaved to generate a base peak ion of *m*/*z* 215.03 and an abundant ion of *m*/*z* 243.07. In addition, the B ring is also a potential fragmentation site due to the presence of the ester bond. The abundant ions *m*/*z* 115.05 and 226.06 backed the rule above because of the cleavage of ring B of F_39_. Compound **39** was designated to be neoflavonoid Ⅷ, and its product ions *m*/*z* 212.0476 and 184.0526 corresponded to the cleavage of the D and B rings, respectively.

## 3. Material and Methods

### 3.1. Material and Reagents

HPLC-grade acetonitrile and methanol were purchased from Ourchem (Shanghai, China). Formic acid (for LC–MS) was purchased for TGI (Tokyo, Japan), and ultrapure water (18.2 MΩ/cm at 25 °C) prepared by a Millipore Alpha-Q purification system (Milipore, Bedford, MA USA). All of the other reagents were analytically pure. The dried heartwoods of *Dalbergia odorifera* T. Chen were collected from Hainan province of China in 2019. The 18 reference standards (purity ≥ 98.0%) were purchased from Shanghai Standard Technology Co., Ltd. (Shanghai, China)., including baicalein (F_1_), apigenin (F_2_), wogonin (F_3_), acacetin (F_4_), galangin (F_5_), quercetin (F_6_), isorhamnetin (F_7_), glycitein (F_11_), genistein (F_12_), calycosin (F_13_), daidzein (F_14_), formononetin (F_15_), isoliquiritigenin (F_20_), naringenin (F_22_), tangeretin (F_23_), liquiritigenin (F_24_), taxifolin (F_28_) and dihydromyricetin (F_29_).

### 3.2. Standard Solution Preparation

Seven reference compounds were mixed and dissolved in 100% methanol to prepare stock solution (approximately 50 μg·mL^−1^ for each compound). The solution was centrifuged at 14,000 rpm for 10 min, and the supernatant was stored at 4 °C before analysis.

### 3.3. Sample Preparation

The dried and powdered heartwood of *D. odorifera* (50 g) was extracted two times with 70% (*v*/*v*) ethanol (total amount 1000 mL) under reflux and then filtered by filter paper. The solution was evaporated under reduced pressure to yield EtOH extract (25 g). The residue was dissolved in 75 mL methanol, and 3 mL of the solution (about 1 g residue) was submitted to column chromatography over silica gel and eluted with a gradient solvent system of petroleum ether–acetone (*v*/*v*, 20:1, 10:1, 8:1, 6:1, 4:1, 2:1, 1:1 and 0:1, each 200 mL) to give eight fractions (Fr1–Fr8). The solutions of eight fractions were evaporated under reduced pressure, and each residue was dissolved in 10 mL methanol. Samples of eight fractions were centrifuged at 14,000 rpm for 10 min, and the supernatant was stored at 4 °C before UHPLC–MS analysis.

### 3.4. UHPLC–LTQ-Orbitrap MS System

The UHPLC separation coupled to MS detection was conducted on an Ultimate^®^ 3000 UHPLC system (Thermo Fisher Scientific, San Jose, CA, USA). An ACQUITY UPLC^®^ HSS T3 column (2.1 × 100 mm, 1.8 μm; Waters, Milford, MA, USA) maintained at 30 °C was used and eluted by a binary mobile phase consisting of 0.1% formic acid (A) and acetonitrile (B). A gradient elution program was set as follows: 0–15 min, 20–28% (B); 15–20 min, 28% (B); 20–30 min, 28–40% (B); 30–40 min, 40–90% (B); 40–45 min, 90% (B); 45–45.1 min, 90–20% (B); and 45.1–50 min, 20% (B). The flow rate and sample injection volume were set at 0.3 mL/min and 1 μL.

A LTQ-Orbitrap Velos Pro hybrid mass spectrometer (Thermo Fisher Scientific, San Jose, CA, USA) equipped with an ESI source was employed to acquire high-resolution MS data in the positive and negative ion mode. The following source parameters were applied: spray voltage, 3.8 kV; capillary temperature, 350 °C; source heater temperature, 300 °C; sheath gas (N_2_), 40 arbitrary units; and auxiliary gas (N_2_), 10 arbitrary units. Two different fragmentation modes, CID and HCD, were employed in data-dependent acquisition mode. Normalized collision energies (NCEs) in CID mode and HCD mode were set at 40% and 80%, respectively. A duty circle included two events (I and II). Full scan over *m*/*z* 240–1200 at a resolution of 30,000 (FWHM defined at *m*/*z* 400) in profile format (p) was performed in Event I. Event II recorded MS2 spectrum of most intense ions. The resolution of Event II was 7500. The minimum signal intensity to trigger MS2 fragmentation was set to 10,000. Dynamic exclusion parameters were set as follows: repeat count, 2; repeat duration, 10 s; exclusion list size, 50; exclusion duration, 10 s; low exclusion mass width, 1.5 Da; and high exclusion mass width, 1.5 Da.

### 3.5. Data Analysis

Data post-processing was performed on the obtained raw data with different software. MSConvert software was used to convert the raw data into mzML files. For MS-DIAL, the mzML files were uploaded for MS features extraction, and the MS/MS positive ion public spectrum database was imported. The other MS-DIAL parameters were as follows: adducts, [M + H]^+^, [M + Na]^+^ and [2M + H]^+^; retention time tolerance, 50 min; accurate mass tolerance (MS1/MS2), 10 mDa; identification score cutoff, 80; sigma window value, 0.5; minimum peak height, 10,000 amplitude; and mass slice width, 0.1 Da. For CD 3.3 software, the raw data were directly imported. The workflow template of metabolomics (Untrg. Metabolomics w Statistics Detect Unknowns w Mapped Pathways (BioCyc beta) and ID, using Online Databases) was selected for data processing. Moreover, on the basis of the system default, the parameters are simply adjusted to get the workflow tree. The parameters were adjusted as follows: adducts, [M + H]^+^, [M + Na]^+^ and [2M + H]^+^; elemental composition, C, H and O; and accurate mass tolerance (MS1/MS2), 10 mDa. For Progenesis QI, the raw data were imported for peak extraction. Four databases were respectively imported for the characterization of unknowns, and the accurate mass tolerance was set to 10 ppm. For SIRIUS 4.9.8, the mzML files were analyzed, and the parameters were set as follows: instrument, orbitrap; MS/MS isotope scorer, score; MS2 MassDev, 10 ppm; candidates, 10; candidates per ion, 1; adducts, [M + H]^+^, [M + Na]^+^ and [2M + H]^+^; elemental composition, C, H and O; use heuristic above *m*/*z*, 300; use heuristic only above *m*/*z*, 650; search in DBs, all; and canopus, parameter-free.

## 4. Conclusions

In this study, an approach that integrates enhanced column separation, intelligent data post-processing and result verification was proposed and successfully applied to the characterization of flavoniods in *D. odorifera*. As a result, 3456 mass features were detected and a total of 197 flavonoids were identified or tentatively characterized, including 7, 80, 95 and 15 compounds in levels 1, 2a, 2b, and 3, respectively. Furthermore, the results of levels 2a–3 were validated by fragmentation rules. Through the above process, unknown compounds in complex matrices were quickly and efficiently identified with existing software bioinformatics tools with maximal identification confidence. It is expected that the discussed method above could be a reference for the analysis of other TCMs.

## Figures and Tables

**Figure 1 molecules-27-02333-f001:**
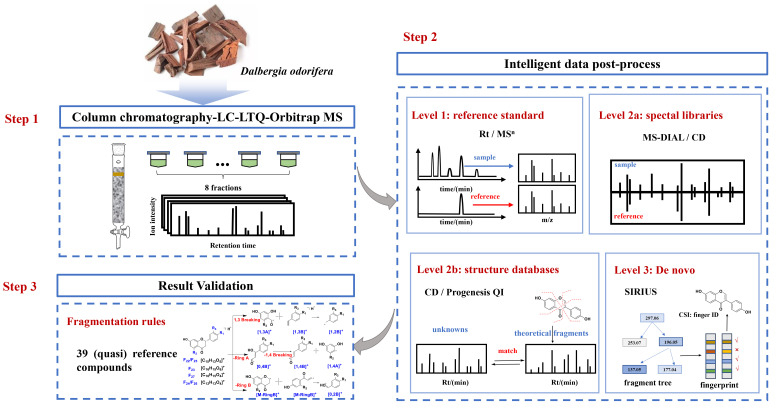
A flowchart for software-assisted multi-tiered mass spectrometry identification of flavonoids in *Dalbergia odorifera*.

**Figure 2 molecules-27-02333-f002:**
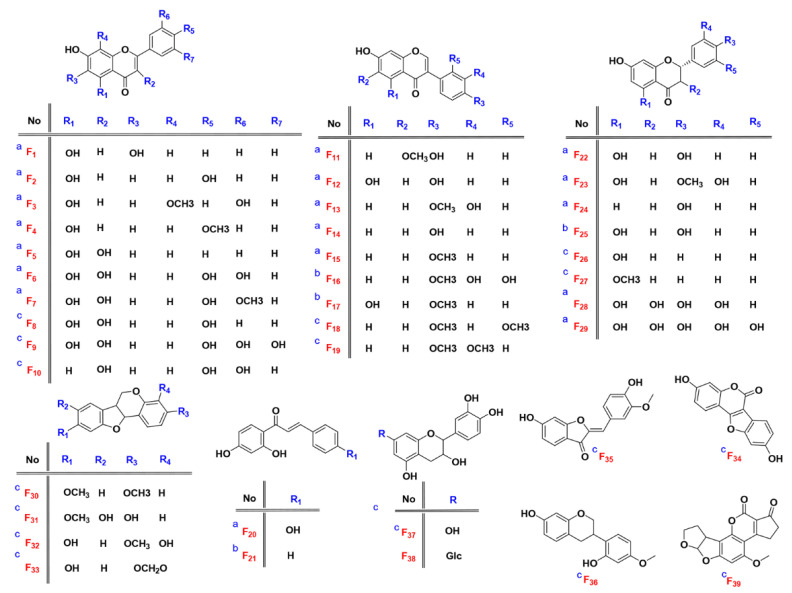
Structures of 39 reference standards and quasi reference standards. (**a**) reference standards; (**b**) quasi reference standards from level 2a; (**c**) quasi reference standards from mzCloud database).

**Figure 3 molecules-27-02333-f003:**
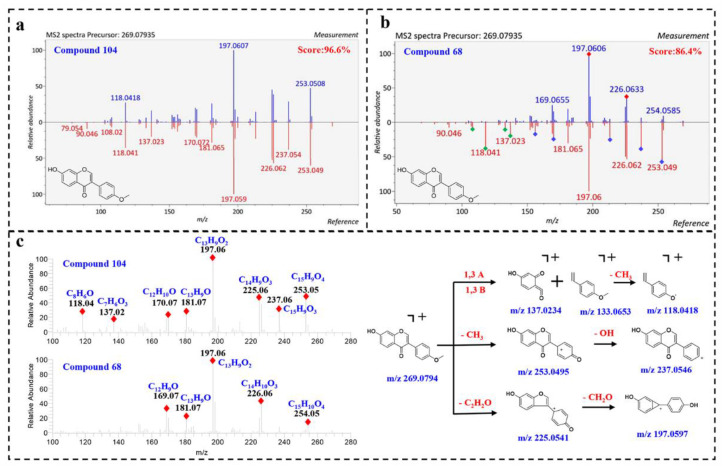
Proposed MS fragmentation pathways of Biochanin B; and matching diagram of compound 104 and 68 in MS-DIAL. (**a**) matching diagram of compound 104 in MS-DIAL; (**b**) matching diagram of compound 68 in MS-DIAL; (**c**) proposed MS fragmentation pathways of Biochanin B.

**Figure 4 molecules-27-02333-f004:**
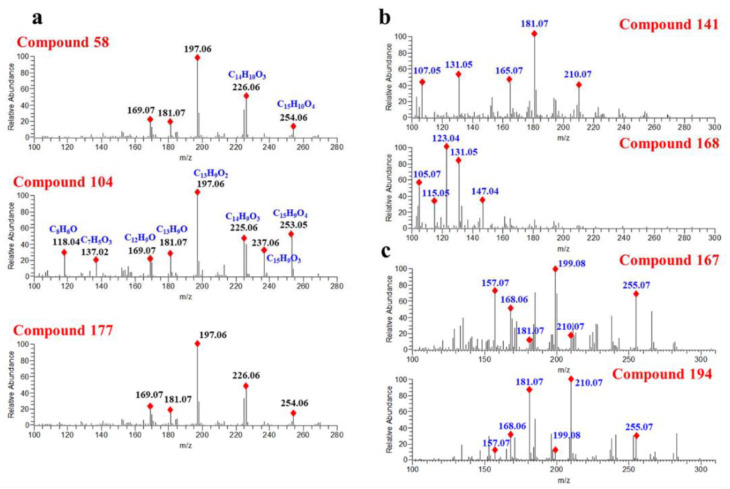
Fragment spectra of compounds annotated in Level 2b: (**a**) compounds annotated by Progenesis QI and (**b**,**c**) compounds annotated by Compound Discoverer.

**Figure 5 molecules-27-02333-f005:**
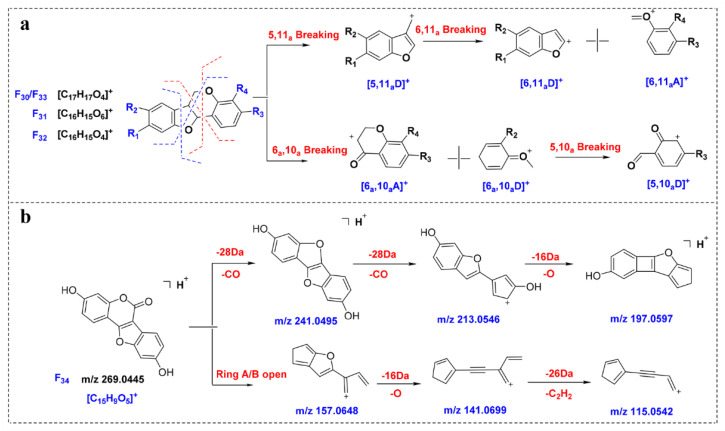
Proposed MS fragmentation pathways of compounds F_30_–F_3__4_. (**a**) F_30_–F_33_; (**b**) F_34_.

**Figure 6 molecules-27-02333-f006:**
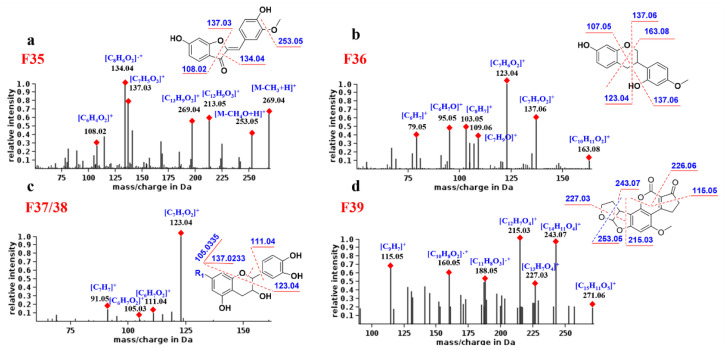
Proposed fragmentation pathways and fragments annotation of F_35_–F_39_. (**a**) F_35_; (**b**) F_36_; (**c**) F_37_ and F_38_; (**d**) F_39_.

## Data Availability

The data presented in this study are available in Appendix A.

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
