# Peer review of "Software Assisted Multi-Tiered Mass Spectrometry Identification of Compounds in Traditional Chinese Medicine: Dalbergia odorifera as an Example"

_molecules, 2022, doi:10.3390/molecules27072333_

Round 1

Reviewer 1 Report

The work by Wang et al., on Software assisted multi-tiered mass spectrometry identification of compounds in TCM: Dalbergia odorifera as an examples was thoroughly investigated and warrants publication to the journal molecules with minor corrections. 

In abstract, line 7, replace is with "was" on line that starts with secondly. On line that starts with finally, replace are with "were" and delete identifications results and replace with "flavonoid compounds". 

In introduction: I encourage to reword line 6 that starts with Although nuclear magnetic resonance.......This line may not be clear to the readers.

Page 6: 2.3.2 CD. Line 8. put "n" on As show........

Conclusion.  replace was with "were" on line 4. 197 flavonoids were..... and please shorten this line. It is long.

The work was thoroughly investigated and smartly presented and warranted publication to the journal with minor corrections.

Reviewer 2 Report

Unambiguous identification of metabolites is challenging due to the complex nature of metabolites, technical limitations, and limited analysis programs. However, mass spectrometry-based metabolite analysis is rapidly growing with a lot of recent developments in the analysis programs. The herbal and medicinal plant-based metabolites are active ingredients that play a vital role in several herbal medicines.  Systematic and confident identification of these active ingredients is crucial for the determination of clinical applications. In the current manuscript, the authors identified and characterized 197 flavonoids from the Dalbergia odorifera plant using a mass spectrometry-based multi-tiered metabolite analysis strategy. The authors isolated the Dalbergia odorifera extract, purified and subjected it to a high-resolution mass spectrometer. The raw data were processed by various analysis tools. In order to reliably identify the metabolites, the authors added reference standards in the sample and analyzed the raw data with a spectral library database on MS-DIAL and Compound Discoverer. The molecular structure was identified by the de novo analysis method. This multi-tiered method is particularly essential in the confident identification of metabolites. The manuscript is well-written, and conclusions are supported by enough data. However, I have a few comments that will help to improve the manuscript.

  1. The authors performed the separation and fractionation of the sample on the silica gel and claimed that they have detected a total of 3456 mass features. It is not very clear how these mass features were detected on silica gel chromatography. There could be enough description of this improved silica gel chromatography in the methods section.
  2. The raw MS data processing is the key to the metabolite identification strategy. The authors mentioned that they used several tools to reliably identify the small molecules but gave very limited information in the section “3.5. Data analysis”. The authors should give enough details about the key steps and parameters used for analysis for each program they used.

Minor point:

  1. Need full name of section “2.3.2. CD”. CD and Compound Discoverer sections are repeated and need renaming.

Reviewer 3 Report

Dear Authors,

Thank you for such interesting and well-orginized manuscript. It is extremely important to develop a new non-target screening methods, however, it should be done carefullly. According to Orbatrap limitations, only 10^6 of ions could be inside analyzer at one moment. Had the Authors checked the influence of concentration to collected results? To provide additional confirmation of the results accuracy, it should be better to show results of spiked with standards samples.
